# An improved multipath video data communication in a vehicular delay-tolerant network

**Khaled H. Almotairi**[ID]*

Computer Engineering Department, Computer and Information Systems College, Umm Al-Qura University, Makkah, Saudi Arabia

* khmotairi@uqu.edu.sa

**Data Availability Statement:** All relevant data is within the paper and its Supporting Information files.

**Funding:** The authors would like to thank the Deanship of Scientific Research at Umm Al-Qura University (https://uqu.edu.sa) for supporting this

## Abstract

A vehicular network offers diverse beneficial services related to video streaming in different types of setups, including rural and urban. Some of the recent issues in vehicular communication include prospect of leveraging machine learning and blockchain for privacy and security enhancement, and resource allocation for video streaming coupled with integration of 6G networks for high data rate. Considering the extreme mobility and dynamic structure of vehicular networks and the high data rates of video streams, a unitary route may not support the required quality of a video stream. To achieve load balancing, connectivity among vehicles, path diversity, and low delay, the multipath transmission with a delay-tolerant network (*DTN*) concept based on a node disjoint algorithm is considered. In this proposed study, video frames are categorized in accordance with priority and forwarded via two graded paths. The first path carries the video reference frame, which is the most important frame for video decoding. The second path carries neighboring frames during video transmission. For the efficient selection of an optimal relay vehicle, a communication cost function is introduced into the existing *DTN*. This communication cost function is based on three key enhancement parameters: link stability rate, accessible bandwidth estimation, and transmission delay. The improvement in this study, is the integration of store-carry-forward strategy to the existing multipath data forwarding strategy. On the basis of the simulation outcomes, the proposed multipath video data communication in a vehicular DTN (MVDTN) scheme can enhance video data delivery in terms of packet loss ratio, end-to-end delay, structural similarity index measure, and peak signal-to-noise ratio. Considering the aforementioned metrics, our proposed schemes outperform the baseline schemes, namely, road-based multi-metrics forwarder selection evaluation for multipath video streaming and quality of service-aware multipath video streaming for an urban vehicular ad hoc network by using ant colony optimization.

## 1. Introduction

Current developments in the domain of intelligent transportation systems have led to the exploration of different techniques, including the delay-tolerant network (*DTN*) approach [1,

work by Grant Code: (22UQU4320277DSR03) to KA. The funders had no role in study design, data collection and analysis, decision to publish, or preparation of the manuscript.

**Competing interests:** The authors have declared that no competing interests exist.

2]. DTN performs better in situations wherein traditional networks fail. DTN is a concept that accepts some level of latency during communication to achieve optimal data delivery. A vehicular DTN (VDTN) applies the concept of conventional DTN [3]. It is a network in which communication is performed with the aid of vehicles as nodes and a roadside unit (RSU) as an immovable node. Important applications of VDTN include on-road accident warning, traffic status notification, vehicle collision prevention, real-time advertisements, and infotainments [4–8]. A vehicular network is characterized by a highly dynamic topology that can be discontinuous in some scenarios. VDTN is frequently used in a challenging network environment wherein the direct path from the source node to the destination node is unknown. A vehicular network takes advantage of this opportunistic concept and adopts the store-carry- forward approach similar to that of DTN [9]. At present, the key challenge in dynamic topology is providing a strategy that can ensure reliable communication. Several proposals have been made in the aspect of VDTN in recent years because VDTN is an efficient approach for handling the disconnection problem in vehicular networks. However, VDTN design still faces challenges, including unpredictable delivery time, incomplete data forwarding, and packet drop due to topology/route change of a vehicular node [10, 11]. Apart from these challenges, multimedia data, such as video files, are frequently large and require a stringent low delay time for efficient delivery. Video streaming services has achieved considerable relevance in wireless communication, including vehicular networks [12, 13]. A vehicular node streams video through mobile terminals, and sophisticated cameras are embedded into vehicles and RSUs. The transmission rate and bandwidth requirements for video streaming are increasing, leading to the use of multipath data forwarding to attain path diversity and concurrent transmission for efficient data delivery [14]. In the multipath approach, video data are segmented into many parts and simultaneously transferred along multiple routes [15]. Considering existing VDTNs, a certain level of latency that is acceptable for non-video data but unacceptable for video data is still experienced. The disconnection of vehicular nodes due to frequent changes in location of the RSU that supplies video data and other streaming vehicles also contributes to latency. High latency in video streaming leads to playback frozen, which can affect the quality experience of users. Thus, minimum delay is required when video streaming in a vehicular network. Meanwhile, multipath forwarding decreases the delay encountered during the store-carry-forward approach in VDTNs. Vehicular node experiences intermittent disconnection due to high mobility thus, requires a strategy such as the store-carry-forward in order to transmit video data reliably. This is because, in store-carry-forward strategy node can store data into its buffer for a certain amount of time until there is connection with another node before data if forwarded. The consideration of multipath transmission is to achieve load balancing since video data are split and can be forwarded via more than one path. Hence, the multipath strategy can be used to reduce the duration of delay in the conventional *DTN*, and reliability of video data delivery can be attained using the store-carry-forward strategy. At the time of writing this paper, there are no studies that considered both multipath and the store-carry-forward strategies for video data transmission in vehicular network. In addition, the estimation of video data communication cost for selection of relay node is essential for choosing most efficient and reliable node for video data transmission. Thus, this has motivated our research, and we propose a multipath video data communication in VDTN (MVDTN). The actual contributions of the present study are as follows:

- Design of the store-carry-forward approach that considers multipath video data transmission.

- Estimation of video data communication cost for relay nodes and path selection based on VDTN.

- Priority-based video frame forwarding that considers multiple paths.

The remaining sections of this paper are structured as follows. In Section 2, related work based on current trends in VDTNs is discussed. The proposed MVDTN scheme is described in Section 3. Section 4 presents the implementation and simulation results with their analyses. Finally, Section 5 provides the conclusions and recommendations of the study.

## 2. Related work

This section presents an extensive review of VDTNs and multipath concepts in video data transmission. VDTN Fu, Zhang, Feng and Zheng [16] aims to augment a category of vehicular network applications; in its nature, this category is characterized by asynchronous data traffic and delay-tolerant data. Such applications do not require end-to-end connectivity and may tolerate a certain level of data loss. VDTN exhibits the concept of asynchronous, end-to-end, and bundle (i.e., variable length message) oriented communication [17]. All these concepts are inherited from conventional DTN. VDTN uses a layered architecture that varies from the DTN architecture, such as the Internet Protocol (IP) based on the VDTN concept and the use of control and data plane parting that considers out-of-band signaling. The disparity between DTN and VDTN network architectures is that the former integrates a variable length layer between the application and transport layer, facilitating the store and forward edge network that enables multiple connections of heterogeneous networks. By contrast, the VDTN architecture incorporates a variable length message into the network layer, presenting an IP over the VDTN strategy. A variable length message can be considered a protocol data unit placed at the VDTN network layer, and it signifies the collection of IP datagrams with similar attributes, e.g., quality of service (QoS), destination node, and application. A distribution storage is based on infrastructure that can partition data across multiple physical servers, which is usually across more than one data center. On the other hand, the store-carry-forward strategy is where each node stores data packets in the buffer, and whenever the node meets at another node, then it forwards the duplicated data packet.

As shown in (S1 Fig), that is, the illustration of video packet forwarding, which includes the I-frame, the P- and B-frames are transmitted via two paths: $pt_1$ and $pt_2$. $pt_1$ carries the I-frame, while $pt_2$ carries the P- and B-frames from the source to the destination. This concept is adopted to achieve load balancing and path diversity for effective video packet delivery.

Meanwhile, recent advancements in the domain of VDTNs should be discussed to explore their limitations and advantages for further improvement. Recent studies have been categorized into multipath oriented video data for vehicular networks (discussed in Subsection 2.1) and delay-tolerant oriented vehicular networks (discussed in Subsection 2.2).

### 2.1. Multipath oriented video data for vehicular networks

Multipath video data forwarding in vehicular networks refers to the transmission of video frames or sub-streams through more than one part to achieve lower traffic load than through a single part and attain load balancing. Overall, the multipath approach helps achieve QoS in video data routing. On the basis of baseline studies, a parallel multipath communication that used forward error correction in video application was proposed in [18]. In this previous work, a recovery approach for loss of actual data with redundant data information was considered. Furthermore, a few suggestions that cover several characteristics of multipath video streaming were presented. These suggestions include storage capacity and delivery latency [8, 19–21], video data replication [22], and buffer enhancement and management [11, 23–25]. An encounter-based routing approach that considers only the node with a maximum degree of

connectivity was recommended in [26]. The node with the highest connectivity guarantees higher video data delivery from the forwarding node to the targeted node. With regard to data replication, an adaptive replica strategy that implements restriction on the amount of packet copies to be stored by an individual node was proposed in [27]. The restriction can be tuned by considering the number of fresh messages received and transmitted by a node. However, the aforementioned studies did not consider DTN for improved video data delivery. In Kumar and Kim [28], a probabilistic trust-centric data replication scheme was considered in VDTN. This scheme adopts five different strategies, namely, replica cost function, data communication cost, data storage cost, data update cost, and trust calculation metrics.

A QoS-aware multipath video data forwarding scheme in urban vehicular networks that applies the ant colony optimization (ACO) algorithm to improve video quality delivery was proposed in [29]. This study used the ACO technique for determining primary and secondary paths and improving the QoS routing path. In addition, the user datagram protocol (UDP) and transmission control protocol (TCP)-expected transmission count have been used for inter-frame and intra-frame transmissions, respectively. To minimize packet loss ratio in multipath routing over an urban vehicular network, optimal video stream distribution via multiple paths that also satisfies QoS parameters was presented in [30]. The considered QoS parameters included the amount of transferred video packets and freezing delay constraints. Thus, the reconstruction and playback of video were attained with assured QoS. Furthermore, a multipath routing strategy for improving QoS in video streaming over a mobile ad hoc network was suggested in [31]. The study used a real-time multimedia application to evaluate two prominent multipath routing protocols: multipath ad hoc on demand distance vector and multipath destination-sequenced distance vector. However, this previous work did not consider high-speed nodes and topology-restricted patterns, such as in the case of vehicular networks. In most multipath video data streaming approaches, VDTN that handles the intermittent disconnection issue has not been considered. DTNs adopt the carry-store-forward technique, which exhibits the capability of handling packet losses due to disconnection problem.

## 2.2. Delay-tolerant oriented vehicular network

Considering the concept of DTN, a relay selection approach that considers multiple hop connection to a targeted vehicle was proposed in [32]. The vehicle with the maximum number of connections to the targeted node is selected as the preferred relay vehicle. Similarly, a relay strategy that investigated network performance and outage gain was suggested for improving data delivery in Feteiha, Hassanein and Kubbar [33]. In this strategy, vehicles are used as bridges/connectors between on-road infrastructure to extend communication coverage. An adaptive carry-store-forward concept that considers two hop communication was proposed for retransmitting missing data in [34]. The retransmission of missing data is performed by the RSU through a relay node that stores data for onward forwarding to the targeted vehicular node. Furthermore, the target vehicle adopts the speed of the relay vehicle for efficient delivery. A cooperative store-carry-forward scheme for reducing outage during data forwarding was suggested in [35]. This scheme uses two directional vehicles as a relay node for the store-carry-forward process through inter-RSU collaboration. The initial relay vehicle is selected by the RSU for data forwarding, and it subsequently forwards the data to the next available RSU. The next available RSU evaluates the suitability of the initial relay vehicle to continue forwarding. If it is unsuitable, then a new relay vehicle is selected for data forwarding. However, the scheme is complex during the collaboration/cooperation phase, which can take more time during relay vehicle selection. In another scheme, an enhanced bivios relay selection concept based on V*DTN* was presented in [17]. This adaptive approach applies the multi-copy-relay concept to

RSU. The RSU uses multiple criteria for selecting multiple vehicles. In this scheme, the relay vehicle is only selected when the target vehicle receives an incomplete data. However, the RSU can only connect to two vehicles at one time, minimizing the possibility of selecting the most efficient vehicle as the relay node. A decision scheme for data forwarding, called the probabilistic bundle relaying scheme, was proposed for improving relay selection. In this scheme, the RSU estimates the suitable vehicle and makes a decision on whether to release the data bundle to a moving vehicle. Suitability estimation and decision-making are determined on the basis of several criteria [36]. Furthermore, density adaptive routing based on a node-aware strategy was presented for enhancing data packet delivery. In this adaptive scheme, epidemic routing is performed when the relay node is located in a low-density area [16]. Moreover, the node determines the importance of a data packet on the basis of a metric known as utility incremental value. A replication-centric strategy, called GeoSpray, which is based on the spray-and-wait procedure, was proposed in [9]. The procedure limits the number of packet copies that can be forwarded by the sending node. In this strategy, the multi-copy concept is adopted during the initial stage and restricted copies of the packets are spread. This strategy offers better delivery possibility compared with some existing schemes, such as Geopps [37], which focus only on multi-copy routing without considering spreading. The preceding studies may be effective in handling the intermittent disconnection problem in a vehicular network; however, they may not efficiently support a high-data-rate network, such as video streaming in a vehicular network. The splitting and distribution of video streams via multiple paths that consider frame priority may minimize packet loss due to high data rates.

## 3. MVDTN

The MVDTN is discussed in this section. The system model and problem formulation are presented in Subsection 3.1, while MVDTN is comprehensively discussed in Subsection 3.2.

### 3.1. Problem formulation and system model

The illustration of the system architecture with its interconnected components is presented in (S1 Fig). As shown in the figure, two paths are determined for forwarding video streams: $pt_1$ and $pt_2$. $pt_1$ forwards a video frame, called the inter-frame or I-frame. $pt_2$ forwards a video frame, called the intra-frame or B- and P-frames. The structure of video frames is depicted in (S2 Fig). Considering the importance of the I-frame, which is typically used to predict the P- and B-frames for the meaningful decoding of a video file, it is given higher priority by storing multiple copies of the I-frame during the store-carry- forward process. The neighboring frames (B- and P-frames) are forwarded via a secondary path, i.e., $pt_2$, which has less priority. Further discussion regarding group of pictures and Moving Picture Experts Group is provided by [38]. Meanwhile, video streaming requires a backbone for continuous access to the video data until complete download/upload of the video file is achieved. Therefore, a strategy for storing video files must be developed to maintain continuous connectivity during video streaming. The time required by a vehicle to exit a network is probabilistic in nature, and it must be considered as an adaptive measure by the vehicular node to regulate network topology. In addition, video streaming is frequently characterized by a high data rate, and thus, it requires more bandwidth for successful transmission from the forwarding vehicle to the target vehicle. Accordingly, splitting a video stream is necessary, and the split video frames are forwarded through multiple paths to achieve path diversity and load balancing during transmission.

Assume a graph $G = (P, E)$ $G = (P, E)$, where $P$ represents the vertices $P = p_1, p_2, \ldots, p_n$ and $E$ the edges $E = e_1, e_2, \ldots, e_n$, such that $\forall P$ indicates an existing homogeneous communication

range. The relay vehicle can only store copies of the I-frame. However, the I-frame is forwarded through the path with the highest performance, which has the lowest cost. Thus, an optimized selection of relay node is implemented for enhanced performance. Therefore, we formulate the cost of selecting a suitable vehicle for video transmission as the data forwarding cost (*DFC*). *DFC* is a function of several parameters, namely, accessible bandwidth (*AB*), link stability rate (*LSR*), and time of data delivery. *AB* is an important parameter that considers a high data rate, such as a video.

The efficient transmission of video streams strongly relies on AB in the communication channel. The medium contention is estimated based on Eq 1.

$$\text{Medium Contention} = probability\,(vp + N + Sp) \tag{1}$$

Medium contention is estimated as the probability of the video packet collision *vp* of a given value *N*, the vehicle with a mean speed *Sp*. The *vp* is considered as the unsuccessful data packet transmitted, which is due to channel medium contention. In other words, it can be estimated as number of packets dropped in the course of data forwarding. The *Sp* is expressed based on Eq 2.

$$Sp = \frac{D_v}{T_E} \tag{2}$$

Available bandwidth is estimated between the present relay vehicle (PRV) and the candidate relay vehicle (CRV). The expression of AB is given in Eq 3.

$$AB_{CRV} = (1 - K) \times (1 - (prob(vp, N, Sp)) \times ID_{rv} \times ID_{tv} \times C, \tag{3}$$

where $ID_{rv}$ and $ID_{tv}$ represent the idle time of the receiving and transmitting vehicles, respectively. The probability of the video packet collision *vp* of a given value *N* the vehicle with a mean speed *Sp*, and the capacity of link *C* are considered for bandwidth estimation. In addition, the back-off estimation mechanism is represented as *K*, and *K* is expressed in Eq 4.

$$K = \frac{DIFS + \overline{back - off}}{D_{diff}} \tag{4}$$

The period variation between the forwarding of two video frames is denoted as $D_{diff}$, while *DIFS* indicates the distributed coordination function inter-frame space. *Back-off* is the mean value of the decreased back-off slot for a specific frame. The $AB_{CRV}$ value is normalized to be between the range of 0 and 1, i.e., $0 \leq AB_{CRV} \leq 1$.

*LSR* relies on the density of the vehicular network and the intensity of mobility. Thus, *LSR* is expressed in Eq 5.

$$LSR_{CRV} = \delta e^{-\alpha t}, \tag{5}$$

where $\delta$ represents the density of vehicles at a relay node area, $\alpha$ is the vehicle speed, and *t* is the time series for each candidate vehicle. In addition, time series is used for different timings to estimate the value of communication cost.

Transmission delay (*TD*) is one of the key parameters for achieving efficient video data delivery, and it can justify the existence of load in a communication channel and vehicle density. The delay for forwarding a packet *q* is determined using Eq 6.

$$TD_L = T_s - T_r, \tag{6}$$

where $T_s$ is the time when a video frame is sent from the relay node, and $T_r$ is the time when a video frame is received at the next forwarding vehicle. The mean delay for a time range *T* is

expressed in Eq 7.

$$TD_{CRV} = \frac{\sum_{q=1}^{N_r} TD_L}{N_r}, \tag{7}$$

where $N_r$ denotes the amount of received frames for the duration of $T$. Further delay difference in an actual interval $T$ is expressed in Eq 8.

$$TD_F = \frac{\sum_{q=1}^{N_r} (DT_S - DT_r)^2}{N_r} \tag{8}$$

Therefore, the communication cost of video data (*CCVD*) can be represented as Eq 9.

$$CCVD = AB_{CRV} + LSR_{CRV} + TD_F \tag{9}$$

By integrating weight values, i.e., $\beta_1 + \beta_2 + \beta_3 = 1$, we derive Eq 10.

$$CCVD = \beta_1 \times AB_{CRV} + \beta_2 \times LSR_{CRV} + \beta_3 \times TD_F \tag{10}$$

Hence, *CCVD* is evaluated for every path before video data transmission.

## 3.2. Information gathering phase

During this phase, which is (S1 Algorithm), the information required for vehicles to communicate is exchanged via a hello message (*HM*). Information from *HM* is used by the present forwarding vehicle or the source vehicle node (*SVN*) to select among candidate next forwarding vehicles (*C-NFVs*). The *HM* format comprises the time stamp for the *HM* receipt, the number of neighbors of a *C-NFV*, and the *HM* counter is set, providing the total number of *HMs* received within every minute from each *C-NFV*. In addition, the position, direction, and speed of a vehicle are also part of the *HM* format and comprise the content of the information table of a neighbor node (*ITN*). An exchange of *HM* occurs every 1s between intermediate nodes, and such exchange is called *HM* packet interval. This phenomenon is realistic because an urban scenario is considered for network settings. Meanwhile, vehicles in both directions are used in video data packet forwarding. All these details are stored in *ITN* and considered in selecting the optimal relay node for video streaming. The information format of *ITN* is depicted in (S3 Fig).

The general assumption in geographic routing states that *SVN* is aware of the location and direction of the destination vehicle node (*DVN*) within the network by using a location service system. The Global Positioning System is used in the current research. Thus, the *DTN* concept is integrated into a greedy algorithm, such that important video data are stored, carried, and forwarded to the relay node to minimize packet loss due to disconnection. In addition, the greedy algorithm is enhanced, enabling it to consider the given cost communication and become aware of vehicles closer to the *DVN*. Considering the high data rate of a video file, multiple route data forwarding is adopted to attain load balancing, low delay, and path diversity.

Considering line-by-line explanation of the (S1 Algorithm), Hello packet is created in Line 1 after which the hello timeout is checked whether it has elapsed or not at Line 2, if it has elapsed then Hello message is created, and *ID* with all other information of the node is inserted into the hello message in Line 3–5. Afterwards, hello message is disseminated with present time and new timeout is created in Line 6 and 7. Then hello message procedure is terminated in Line 8 and 9. A fresh procedure for Hello message receive is triggered in Line 10 and 11. It checked whether hello message is received or not, if received the system checks whether the *ID*

is presents in the *ITN* of the receiving node at Lines 12 and 13. Once the node *ID* is not present in *ITN*, new record is created that contains all details of the senders' node in Lines 14 and 15. Otherwise, *ID* and all information of sender node is updated in the *ITN* in Lines 17 and 18. The *HM* is discarded at Line 21, after which the procedure is terminated.

### 3.3. Data forwarding phase

From (S2 Algorithm), video packet forwarding that uses communication cost in relay node selection is discussed. Video frames, including I-, B-, and P-frames, with their corresponding frame *IDs* are initiated in Lines 1 and 2. The algorithm applies *DTN* and communication cost concepts when forwarding video data. The store-carry- forward strategy is used to minimize frame loss due to network disconnection. Communication cost evaluates all intermediate nodes labeled as *C-NFV* from which relay nodes are selected. For *CCVD*, three important metrics, namely, *AB*, *LSR*, and *TD*, are introduced into the video data forwarding algorithm to realize an optimal selection of relay node/NFV. ID of the intended destination node is inserted in the video packet and the forwarding of video packet from source to destination node is triggered in Lines 3 and 4, respectively. Lines 6 to 12 computes the metrics and then estimate *CCVD* for each of *C-NFV* in the network. Thereafter, the first two suitable nodes that exhibit a connection with the *SVN* are selected as relay nodes for multiple routes forwarding in Line 13. The *IDs* of the two chosen relay nodes are inserted in the *ITN* at Line 14. Otherwise, *C-NFV* is not selected as the relay vehicle, thus, the forwarding vehicle *ID* is deleted in Lines 15 and 16. The condition for the best cost is tested on each candidate node, after which the I-frame video packet is stored in the memory of the relay node in Lines 17 and 18. Communication cost is evaluated when selecting the optimal node. The intermediate vehicle with the minimum cost is preferred. The same weighting factor is assigned to the three metrics to allocate equal priority to the metrics. The I-frame is transmitted via the path with the highest minimum cost and other video frames including P-frame and B-frame are forwarded through second highest minimum cost in Lines 19 and 20. The I-frame is carried for a random amount of time before being retransmitted in the case of a packet drop in Lines 21 and 22. The retransmission guarantees the delivery of the important frame, thus, improving the quality of video streaming. Whenever the relay vehicle receives video packet based on Line 25, *CCVD* is computed to select other intermediary vehicles, and data packets are forwarded in Lines 25 to 28. Subsequently, the details of successful and suitable nodes are stored in *ITN* in Line 29. It is further checked whether the candidate node is optimal in terms of minimum cost, when the node is not optimal then the detail of the vehicle is deleted based on Lines 32 and 33. Otherwise, video packet is forwarded to the suitable candidate vehicle in Line 26, hence, data packet forwarding is continuously performed until one of the *C-NFV*s is identified as the *DVN* in Lines 37 and 38. Then, transmission terminates at the target node. The destination vehicle receives a video frame, compares it, and then deletes duplicate I-frames on the basis of video packet *ID* in Lines 39 and 41. The procedure terminates when the *DVN* for optimal video delivery is found in Lines 42 to 44. The flowchart of the video data forwarding procedure is presented in (S4 Fig). It highlights the steps and connections of the video communication process that considers the communication cost function.

## 4. Performance evaluation

The performance assessment of the proposed scheme is conducted through computer simulation in this section. The proposed MVDTN scheme is benchmarked against state-of-the-art schemes, including road-based multi-metrics forwarder selection evaluation for multipath video streaming (*RMF-MVS*) [39] and QoS-aware multipath video streaming for urban

VANETs using *ACO* (QoSM-ACO) [29]. The proposed MVDTN communication process is implemented using the well-known network simulator version 2.3 (NS-2.3) developed by Chen, Wang, Tsai, Chang, Liu, Guo, Lien, Sum and Hung [40]. The mimicking of vehicle movement is performed using the simulation of urban mobility tool (SUMO) of Krajzewicz, Hartinger, Hertkorn, Mieth, Ringel, Rössel and Wagner [41]. SUMO uses a mobility model generator for a vehicular network called MOVE. To generate video frames, the popular Evalvid tool developed by Klaue, Rathke and Wolisz [42] is used. Evalvid also provides the video quality evaluation framework. For the simulation area, Manhattan City's electronic map, which has a longitude of −96.574 to −96.563 and a latitude of 39.191 to 39.184 is considered for traffic and mobility environment configuration as shown in (S5 Fig). The electronic map topology and data are obtained from OpenStreetMap contributors.

The summary of information regarding the simulation parameters are provided in Table 1. This information includes urban simulation region, simulation duration, vehicle speed, number of vehicles, media access control (MAC) protocol used, video strength, video streaming duration, signal coverage, frequency of bandwidth used, propagation model, antenna system, traffic type, channel type, transmission protocol, hello packet time-out, scenario time-out, comparison schemes, and evaluation metrics used.

The performance evaluation of MVDTN is performed on the basis of two circumstances: vehicle density and video packet sending rate (packet per seconds). The performance metrics used for evaluation comprises packet loss ratio (PLR), end-to-end delay (E2ED), structural similarity index measure (SSIM), and peak signal-to-noise ratio (PSNR). PLR is estimated on the basis of the ratio of video packets dropped to the main quantity of video packets forwarded from the source vehicle to the destination vehicle [43, 44]. E2ED is the overall time consumed for a video packet to be forwarded from the source vehicle to the destination vehicle [45, 46]. SSIM is used to calculate the apparent similarity between the main video image and that of the image received after transmission [47, 48]. PSNR indicates the ratio for the highest value of a signal and the degree of distortion that disturbs video quality [23, 49].

**Table 1. Simulation parameters.**

| Parameters | Values |
|---|---|
| Urban simulation area | $1000 \times 1000 \text{ m}^2$ |
| Simulation time | 600 s |
| Vehicle speed | 2.78–13.89 m/s (10–50 km/h) |
| Number of vehicles | 50 to 500 |
| MAC protocol | IEEE 802.11p |
| Video resolution | $352 \times 288$ |
| Video play duration | 139 s |
| Transmission range | 250 m |
| Frequency bandwidth | 5.9 GHz |
| Propagation model | Shadowing |
| Antenna model | Omni-directional |
| Traffic type | Constant bit rate |
| Channel type | Wireless |
| Transmission protocol | TCP and UDP |
| Hello packet time-out | 1 s |
| Scenarios | High-density urban scenario |
| Comparison protocol | RMF-MVS and QoSM-ACO |
| Metrics | PLR, E2ED, SSIM, and PNSR |

## 4.1. Video packet sending rate

In this subsection, MVDTN is evaluated against RMF-MVS and QoSM-ACO on the basis of the PLR, E2ED, SSIM, and PSNR metrics by considering various video packet sending rates. In (S6 Fig), the PLR of the video data that considers various video packet sending rates is presented. In this figure, the number of vehicles is kept constant at 300, which is above the average number of total vehicular nodes considered in the current study. A total of 300 vehicles are considered because an urban scenario is used in this evaluation. The results indicate that PLR increases as video packet sending rate increases. Video packet loss is minimal when data packet sending rate is 1 to 3. The packet loss increases as the number of packets sending rate increases. This is in connection with the fact that there is frequent signaling during communication between all neighbor vehicles. In video streaming, the delivery of data packets with few losses is more important than the delay of ≤5 seconds experienced. Because the loss of packets affects quality of the video streaming. Thus, there is balance in the tradeoff between cost and performance. An increase in the PLR can be related to the high queuing rate at the receiving and sending vehicular nodes and the intermittent disconnection between communicating vehicles. However, packet loss is reduced due to the store-carry-forward strategy. The TD of the video packet is also considered when selecting the most suitable vehicle as the relay node in MVDTN, this assists in the selection of optimal performance. Therefore, the proposed MVDTN outperforms the two baseline schemes, i.e., RMF-MVS and QoSM-ACO, with performance percentage differences of 16.7% and 11.4%, respectively. Table 2 depicts the average values of data packet loss ratio for the proposed MVDTN against the two benchmarking schemes namely, RMF-MVS and QoSM-ACO.

In (S7 Fig), PLR is presented against varying vehicle densities. The video packet sending rate is kept constant at six packets per seconds. The PLR of the video data decreases as the density of vehicles increases. This finding may be related to the improved connectivity approach between vehicles during communication. The improved connectivity approach is based on LSR and the store-carry-forward process for important video frames (I-frames). RMF-MVS and QoSM-ACO have higher PLR compared with the proposed MVDTN scheme because the link breakage issue that leads to packet loss is addressed via the *DTN* concept and the proposed communication cost, namely, transmission delay, and bandwidth availability. Thus, MVDTN outperforms RMF-MVS and QoSM-ACO with performance percentage differences of 13.2% and 9.3%, respectively. Table 3 shows the average values of data packet loss for the proposed MVDTN against the two benchmarking schemes namely, RMF-MVS and QoSM-ACO.

In (S8 Fig), E2ED is plotted against various video packet sending rates. The graph shows an increase in delay as video packet sending rate increases. Probably, this is due to the connection

**Table 2. Average results for video packet loss ratio.**

| Video packet sending rate (packet/sec) | RMF-MVS | QoSM-ACO | MVDTN |
|---|---|---|---|
| 1 | 0 | 0 | 0 |
| 2 | 0.5 | 0.3 | 0 |
| 3 | 0.7 | 0.5 | 0 |
| 4 | 1.0 | 0.8 | 0.3 |
| 5 | 1.3 | 1.1 | 0.6 |
| 6 | 1.7 | 1.5 | 0.8 |
| 7 | 2.2 | 2.0 | 1.0 |
| 8 | 2.7 | 2.4 | 1.3 |
| 9 | 3.3 | 2.9 | 1.6 |
| 10 | 3.9 | 3.5 | 2.0 |

**Table 3. Average results for video packet loss considering vehicle densities.**

| Vehicle density | RMF-MVS | QoSM-ACO | MVDTN |
|---|---|---|---|
| 50 | 13.0 | 11.0 | 5.0 |
| 100 | 9.0 | 8.0 | 4.5 |
| 150 | 7.0 | 6.0 | 4.0 |
| 200 | 5.5 | 5.0 | 3.4 |
| 250 | 5.0 | 4.5 | 3.0 |
| 300 | 4.8 | 4.2 | 2.6 |
| 350 | 4.5 | 3.8 | 2.2 |
| 400 | 4.0 | 3.4 | 2.0 |
| 450 | 3.5 | 3.1 | 1.6 |
| 500 | 3.0 | 2.8 | 1.1 |

with queuing delay at the receiving vehicular node. However, this phenomenon is minimized in the proposed scheme by using the store-carry-forward method when selecting the optimal node. In addition, the TD of various vehicular nodes at the multipath is estimated before selecting the node for data packet forwarding. This procedure assists in achieving lower delay throughout the video data streaming process. The percentage delay is estimated on the basis of the maximal allowable delay of 5s. The proposed MVDTN performs better than RMF-MVS and QoSM-ACO in terms of lower delay with performance percentage differences of 1.8% and 2.5%, respectively. Table 4 shows the average values of data packet end-to-end delay of the proposed MVDTN against the two benchmarking schemes namely, RMF-MVS and QoSM-ACO.

In (S9 Fig), E2ED is plotted against different vehicle densities. The graph depicts a decrease in delay as vehicle density increases. Thus, both QoSM-ACO and MVDTN have lower delay compared to RMF-MVS. This could be due to the congestion in communication with the numerous vehicular nodes that serve as relay nodes with an insufficient store-carry-forward strategy, reducing store-carry-forward delay for each video packet during transmission. In addition, the TD of various vehicular nodes at the multipath is estimated before selecting the node for data packet forwarding. This procedure helps in decreasing delay during video data streaming. The percentage delay is estimated on the basis of the maximal allowable delay of 5s. The proposed MVDTN performs better than RMF-MVS and QoSM-ACO in terms of lower delay with performance percentage differences of 1.1% and 1.6%, respectively. Table 5 depicts the average values of data packet end-to-end delay with respect to vehicle density of the proposed MVDTN against the two benchmarking schemes namely, RMF-MVS and QoSM-ACO.

**Table 4. Average results for end-to-end delay considering data rate.**

| Video packet sending rate (packet/sec) | RMF-MVS | QoSM-ACO | MVDTN |
|---|---|---|---|
| 1 | 0.15 | 0.17 | 0.10 |
| 2 | 0.19 | 0.21 | 0.12 |
| 3 | 0.22 | 0.23 | 0.14 |
| 4 | 0.23 | 0.23 | 0.16 |
| 5 | 0.24 | 0.25 | 0.16 |
| 6 | 0.25 | 0.255 | 0.16 |
| 7 | 0.25 | 0.26 | 0.17 |
| 8 | 0.26 | 0.27 | 0.18 |
| 9 | 0.26 | 0.28 | 0.19 |
| 10 | 0.27 | 0.31 | 0.21 |

**Table 5. Average results for end-to-end delay considering vehicle density.**

| Video packet sending rate (packet/sec) | RMF-MVS | QoSM-ACO | MVDTN |
|---|---|---|---|
| 50 | 0.24 | 0.26 | 0.20 |
| 100 | 0.22 | 0.24 | 0.19 |
| 150 | 0.21 | 0.22 | 0.18 |
| 200 | 0.20 | 0.21 | 0.17 |
| 250 | 0.195 | 0.20 | 0.17 |
| 300 | 0.19 | 0.20 | 0.17 |
| 350 | 0.19 | 0.20 | 0.16 |
| 400 | 0.19 | 0.20 | 0.16 |
| 450 | 0.19 | 0.20 | 0.16 |
| 500 | 0.19 | 0.20 | 0.15 |

(S10 Fig) presents SSIM index on the basis of various video packet sending rates. The various packet sending rate is used to demonstrate that the proposed scheme can carry out fast selection of relay node. The SSIM of video data increases with a slight difference as video packet sending rate increases for all the values considered. The increase in the value of the SSIM index reaches its top when the data rate is above 8packets/sec. The SSIM index value continues to increase when the data rate is above 10packets/sec. Interestingly, the SSIM values for all three schemes are above the average acceptable value. The number of vehicles is 300, which is kept constant during the simulation. RMF-MVS and QoSM-ACO have lower SSIM than the proposed MVDTN possibly due to the relationship between the improved LSR concept and the distributed video frame forwarding concept used for effective video data packet delivery. Thus, the structure of the transmitted video image is nearly the same as the original image before transmission. Therefore, MVDTN outperforms the two baseline schemes, i.e., RMF-MVS and QoSM-ACO, with performance percentage differences of 7.1% and 3.6%, respectively. Table 6 shows the average values of SSIM index with respect to different data rate of the proposed MVDTN against the two benchmarking schemes namely, RMF-MVS and QoSM-ACO.

(S11 Fig) depicts the SSIM index based on different number of vehicles. The structural quality index of video data increases with a slight difference as video packet sending rate increases for all the values considered. Interestingly, the SSIM values for all the three schemes are above the average acceptable value. The number of video packets being sent is six, which is kept constant during the simulation. RMF-MVS and QoSM-ACO have lower SSIM values compared with the proposed MVDTN probably due to the relationship between the improved link

**Table 6. Average results for SSIM index considering various data rate.**

| Vehicle density | RMF-MVS | QoSM-ACO | MVDTN |
|---|---|---|---|
| 1 | 0.820 | 0.840 | 0.860 |
| 2 | 0.810 | 0.850 | 0.865 |
| 3 | 0.800 | 0.840 | 0.868 |
| 4 | 0.805 | 0.838 | 0.880 |
| 5 | 0.820 | 0.850 | 0.885 |
| 6 | 0.830 | 0.860 | 0.889 |
| 7 | 0.831 | 0.860 | 0.892 |
| 8 | 0.831 | 0.860 | 0.897 |
| 9 | 0.832 | 0.860 | 0.897 |
| 10 | 0.832 | 0.870 | 0.910 |

**Table 7. Average results for SSIM index considering various vehicle densities.**

| Vehicle density | RMF-MVS | QoSM-ACO | MVDTN |
|---|---|---|---|
| 50 | 0.780 | 0.800 | 0.856 |
| 100 | 0.810 | 0.820 | 0.860 |
| 150 | 0.800 | 0.810 | 0.850 |
| 200 | 0.815 | 0.830 | 0.870 |
| 250 | 0.810 | 0.820 | 0.865 |
| 300 | 0.820 | 0.840 | 0.870 |
| 350 | 0.831 | 0.843 | 0.880 |
| 400 | 0.829 | 0.839 | 0.892 |
| 450 | 0.831 | 0.845 | 0.900 |
| 500 | 0.830 | 0.843 | 0.890 |

stability concept and the distributed video frame forwarding concept adopted for effective video data packet delivery. Thus, the structure of the forwarded video image is nearly the same as the main image before transmission. Therefore, MVDTN outperforms the two baseline schemes, i.e., RMF-MVS and QoSM-ACO, with performance percentage differences of 6.4% and 5.1%, respectively. Table 7 depicts the average values of SSIM index with respect to vehicle density of the proposed MVDTN against the two benchmarking schemes namely, RMF-MVS and QoSM-ACO.

In (S12 Fig), PSNR that considers different video packet sending rates is presented. When the video packet is one, PSNR starts at its peak but decreases as video packet sending rate increases. The PSNR at when the packet sending rate is 3–9 packet/sec continue to decrease rapidly as the packet sending rate increase. Meanwhile, the number of vehicles is 300, which is kept constant during the simulation. PSNR continues to decrease, but it is still greater than the average value of 20 dB, which is an acceptable value for various video packet sending rates. The improved performance may be related to the splitting of video frames and forwarding them via multiple paths. This step helps in minimizing congestion that can lead to noise in video data. *CCVD* also assist in selecting the optimal node for video data packet transmission. Different parameters, including link stability and AB estimation, are introduced into the greedy and MVDTN routing approach. RMF-MVS and QoSM-ACO have lower PSNR values than the proposed MVDTN probably because both baseline schemes do not consider the MVDTN strategy in the selection of the optimal node for relay vehicles. Therefore, the proposed MVDTN outperforms RMF-MVS and QoSM-ACO with performance percentage differences of 7.3% and 5.1%, respectively. Table 8 depicts the average values of PSNR with

**Table 8. Average results for PSNR considering various data rate.**

| Video packet sending rate (packet/sec) | RMF-MVS | QoSM-ACO | MVDTN |
|---|---|---|---|
| 1 | 33.0 | 33.5 | 35.0 |
| 2 | 31.0 | 32.0 | 34.0 |
| 3 | 30.0 | 31.0 | 33.0 |
| 4 | 28.0 | 29.0 | 31.0 |
| 5 | 27.0 | 28.0 | 30.0 |
| 6 | 26.0 | 27.0 | 29.0 |
| 7 | 25.8 | 26.0 | 28.0 |
| 8 | 25.6 | 25.8 | 27.0 |
| 9 | 25.4 | 25.7 | 26.0 |
| 10 | 25.1 | 25.4 | 25.7 |

**Table 9. Average results for PSNR considering various vehicle densities.**

| Vehicle density | RMF-MVS | QoSM-ACO | MVDTN |
|---|---|---|---|
| 50.0 | 28.0 | 29.0 | 30.0 |
| 100 | 32.0 | 33.0 | 34.0 |
| 150 | 34.0 | 34.5 | 36.0 |
| 200 | 32.0 | 33.5 | 35.0 |
| 250 | 30.0 | 32.0 | 34.0 |
| 300 | 29.0 | 30.0 | 32.0 |
| 350 | 28.0 | 30.0 | 31.0 |
| 400 | 28.0 | 29.0 | 31.0 |
| 450 | 27.5 | 28.5 | 29.5 |
| 500 | 26.5 | 27.5 | 29.0 |

respect to data rate of the proposed MVDTN against the two benchmarking schemes namely, RMF-MVS and QoSM-ACO.

In (S13 Fig), PSNR is depicted against different vehicle densities. PSNR increases as vehicle density increases. The PSNR of video data begins to decrease when the number of vehicles is greater than 150. PSNR continues to decrease, but it maintains an acceptable value that is greater than the average value of PSNR, i.e., 20 dB for all the considered vehicle densities. The video packet sending rate is six, which is kept constant. Better performance is achieved in terms of PSNR despite the high density of vehicles being connected to the communication cost adopted in this study. The strategy uses link stability between vehicles and considers the accessibility of the bandwidth channel in case of congested communication in a vehicular network. In addition, the store-carry-forward concept is used because the *DTN* approach is considered with a modification in the aspect of the nature of the video frame to be transmitted. RMF-MVS and QoSM-ACO have lower PSNR compared with the proposed MVDTN possibly because the baseline schemes do not consider the store-carry-forward concept in relay node operation. Therefore, the proposed MVDTN outperforms RMF-MVS and QoSM-ACO with performance percentage differences of 8.1% and 4.5%, respectively. Table 9 demonstrates the average values of PSNR with respect to different vehicle densities of the proposed MVDTN against the two benchmarking schemes namely, RMF-MVS and QoSM-ACO.

## 5. Conclusions and future work

MVDTN is proposed and implemented to evaluate its performance against baseline schemes, namely, RMF-MVS and QoSM-ACO. The overall objective of the current study is to design and develop a *DTN* in vehicular settings by considering the multipath approach. A communication cost strategy for selecting an optimal relay vehicle is developed for efficient video data delivery. In addition, distributive video forwarding across two paths considers the importance of video frames, with priority being assigned during data forwarding. The store-carry- forward approach in *DTN*s is used with modification in terms of the frames stored and forwarded for efficient video data packet delivery. The simulation results show that the proposed scheme outperforms the baseline approaches. To further improve the proposed scheme in the future, video streaming on the Internet of vehicles that considers artificial intelligence can be explored to enhance the field of autonomous vehicles further.

## Supporting information

**S1 Fig. Video packet forwarding in VDTN.**
(DOCX)

**S2 Fig. Video frames with compression scalability.**
(DOCX)

**S3 Fig. Hello packet format.**
(DOCX)

**S4 Fig. Video data forwarding flowchart.**
(DOCX)

**S5 Fig. Simulation area based on Manhattan city map.**
(DOCX)

**S6 Fig. PLR based on different video packet sending rates.**
(DOCX)

**S7 Fig. PLR based on different vehicle densities.**
(DOCX)

**S8 Fig. E2ED based on different video packet sending rates.**
(DOCX)

**S9 Fig. E2ED based on different vehicle densities.**
(DOCX)

**S10 Fig. SSIM Index based on different video packet sending rates.**
(DOCX)

**S11 Fig. SSIM index based on different vehicle densities.**
(DOCX)

**S12 Fig. PSNR based on different video packet sending rates.**
(DOCX)

**S13 Fig. PSNR based on different vehicle densities.**
(DOCX)

**S1 Algorithm. Information gathering phase.**
(DOCX)

**S2 Algorithm. Video data forwarding based on communication cost.**
(DOCX)

## Author Contributions

**Conceptualization:** Khaled H. Almotairi.

**Data curation:** Khaled H. Almotairi.

**Formal analysis:** Khaled H. Almotairi.

**Investigation:** Khaled H. Almotairi.

**Methodology:** Khaled H. Almotairi.

**Project administration:** Khaled H. Almotairi.

**Resources:** Khaled H. Almotairi.

**Software:** Khaled H. Almotairi.

**Supervision:** Khaled H. Almotairi.

**Validation:** Khaled H. Almotairi.

**Visualization:** Khaled H. Almotairi.

**Writing – original draft:** Khaled H. Almotairi.

**Writing – review & editing:** Khaled H. Almotairi.

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
