## [Decision Letter · Decision Letter 0]

7 Dec 2021

PONE-D-21-30574An Improved Multipath video data communication in Vehicular Delay Tolerant NetworkPLOS ONE

Dear Dr. Almotairi,

Thank you for submitting your manuscript to PLOS ONE. After careful consideration, we feel that it has merit but does not fully meet PLOS ONE’s publication criteria as it currently stands. Therefore, we invite you to submit a revised version of the manuscript that addresses the points raised during the review process.

We look forward to receiving your revised manuscript.

Kind regards,

Jian Shen

Academic Editor

PLOS ONE

Journal Requirements:

Please include your amended statements within your cover letter; we will change the online submission form on your behalf."

Reviewers' comments:

Reviewer's Responses to Questions

**Comments to the Author**

1. Is the manuscript technically sound, and do the data support the conclusions?

Reviewer #1: Partly

Reviewer #2: Yes

Reviewer #3: Partly

2. Has the statistical analysis been performed appropriately and rigorously? 

Reviewer #1: No

Reviewer #2: Yes

Reviewer #3: N/A

3. Have the authors made all data underlying the findings in their manuscript fully available?

Reviewer #1: No

Reviewer #2: Yes

Reviewer #3: Yes

4. Is the manuscript presented in an intelligible fashion and written in standard English?

Reviewer #1: Yes

Reviewer #2: Yes

Reviewer #3: Yes

5. Review Comments to the Author

Reviewer #1: In this paper, the multipath transmission with a delay-tolerant network concept based on a node disjoint algorithm is investigated, where the first path carries the video reference frame and the second one carries neighboring frames during video transmission. A communication cost function, based on link stability rate, accessible bandwidth estimation, and transmission delay, is introduced into the existing delay-tolerant network. The simulations are provided to illustrate the effectiveness of the proposed scheme.

The following comments might be helpful for improving the presentations:

1) The contribution and motivation of this paper are not clear. What is the motivation to design the store-carry-forward with the consideration of multipath video data transmission? What are the advantages of estimating the video data communication cost for relay nodes? The authors are invited to give a detailed discussion on these issues.

2) The performance comparison is presented based on the simulation results. It is insufficient to show the superiority of the proposed MVDTN scheme by simulation. Some experiment results should be given to show the implementability of the designed MVDTN.

3) This manuscript looks like a technical manual rather than a scientific article. The authors should give an intensive theoretical analysis on the proposed results.

4) The readability should be improved of this paper should be improved. There are some typos throughout the paper that should be corrected.

Summing up the above comments, the manuscript can be accepted provided that the above comments are properly taken into account.

Reviewer #2: This paper studies how to transmit video efficiently without losing frames in vehicular network. The problem selection has practical significance. The language of the article is fluent,and the experiment is sufficient. But the innovation is not strong. Can you discuss what is the essential difference between and distributed storage?

Reviewer #3: In this paper, the author proposed a storage-forward method for multipath video data transmission and introduced a communication cost function into the existing delay-tolerant network (DTN).

The whole paper is finished with some problems. There is a long way to go before it can be accepted. We point out the specific problems as follows:

1. First of all, the title of this paper is “An Improved Multipath video data communication in Vehicular Delay Tolerant Network”. From the author's abstract section, it is difficult to see where the improvements proposed by the author are. In fact, the author doesn’t give the current research hot issues in the first few sentences of the abstract.

2. The author's contribution is too few and the description is unclear. It is difficult for me to understand the author's second and third contribution points as two complete ones.

3. In the related work section, the author's description is too redundant. In fact, the related work is a summary of previous work. It is commendable that the author selected several points in this part and cited a large number of literatures for analysis. However, some of the author's concluding remarks are too redundant. For example, on page 6, the author explains some problems in literature [29] in more than 10 lines which is too detailed and unnecessary.

4. The author made a lot of grammatical and formatting errors in his article. For example, there is nothing after the 'and' of the author's contribution; As on page 6, lines 151 and 154, the author quotes [?]. On Line 217 of page 9, the meaning of “Variable f represents” is unknown. The author had better check the whole paper carefully to correct grammatical errors, which appeared too many times and even affected reading.

6. PLOS authors have the option to publish the peer review history of their article (what does this mean?). If published, this will include your full peer review and any attached files.

Reviewer #1: No

Reviewer #2: No

Reviewer #3: No

---

## [Author Response · Author response to Decision Letter 0]

20 Feb 2022

Manuscript ID: PONE-D-21-30574 

Paper Title: Improved Multipath video data communication in Vehicular Delay Tolerant Network. 

To: PLOS ONE Editor

Re: Response to Reviewers

Dear Editor,

Thank you for allowing a resubmission of our manuscript, with an opportunity to address the reviewers’ comments.

We are uploading:

(1) Our point-by-point response to the comments (below) (response to reviewers). 

(2) An updated manuscript with Red Font Colour indicating changes, and 

(3) A clean updated manuscript without highlights.

(4) I have updated the information in the file "Financial Disclosure section" as you asked in your email as follow:

(5) Also, as per your request, I added the below statement in the cover letter file. 

Funding:

The authors would like to thank the ‎Deanship of Scientific Research at Umm Al-Qura University (https://uqu.edu.sa) ‎for supporting this work by Grant Code: (22UQU4320277DSR03) to KA. The funders had no role in study design, data collection and analysis, decision to publish, or preparation of the manuscript.

Best regards,

Dr. Khaled H Almotairi

Email: khmotairi@uqu.edu.sa

---

## [Decision Letter · Decision Letter 1]

17 May 2022

PONE-D-21-30574R1Improved multipath video data communication in Vehicular Delay Tolerant NetworkPLOS ONE

Dear Dr. Almotairi,

Thank you for submitting your manuscript to PLOS ONE. After careful consideration, we feel that it has merit but does not fully meet PLOS ONE’s publication criteria as it currently stands. Therefore, we invite you to submit a revised version of the manuscript that addresses the points raised during the review process.

We look forward to receiving your revised manuscript.

Kind regards,

Jian Shen

Academic Editor

PLOS ONE

Journal Requirements:

Reviewers' comments:

Reviewer's Responses to Questions

**Comments to the Author**

1. If the authors have adequately addressed your comments raised in a previous round of review and you feel that this manuscript is now acceptable for publication, you may indicate that here to bypass the “Comments to the Author” section, enter your conflict of interest statement in the “Confidential to Editor” section, and submit your "Accept" recommendation.

Reviewer #1: (No Response)

Reviewer #3: (No Response)

Reviewer #4: (No Response)

2. Is the manuscript technically sound, and do the data support the conclusions?

Reviewer #1: (No Response)

Reviewer #3: Yes

Reviewer #4: Yes

3. Has the statistical analysis been performed appropriately and rigorously? 

Reviewer #1: (No Response)

Reviewer #3: Yes

Reviewer #4: (No Response)

4. Have the authors made all data underlying the findings in their manuscript fully available?

Reviewer #1: (No Response)

Reviewer #3: Yes

Reviewer #4: (No Response)

5. Is the manuscript presented in an intelligible fashion and written in standard English?

Reviewer #1: (No Response)

Reviewer #3: Yes

Reviewer #4: Yes

6. Review Comments to the Author

Reviewer #1: The paper has been improved. All the previous suggestions have been well considered and thoroughly implemented in this revised paper. Hence, this paper could be accepted for publication now.

Reviewer #3: After the revision of the original version, the following problems still exist in this paper:

1.This paper involves many algorithms, but the part of the derivation is too concise. I think it is necessary to explain the algorithms properly, which can help readers better understand them. It is suggested to give detailed and clear explanation.

2.There are some wrong fonts in the paper. For example, on page 10, line 257, "HM" should be italicized. Please check the whole paper and correct similar mistakes carefully.

3.The format of the references is not uniform in the paper. For example, in references 19 and 32, "IEEE" is lowercase while it is uppercase in other references. The authors should modify the format of references carefully.

After the revision of the original version, the following problems still exist in this paper:

1.This paper involves many algorithms, but the part of the derivation is too concise. I think it is necessary to explain the algorithms properly, which can help readers better understand them. It is suggested to give detailed and clear explanation.

2.There are some wrong fonts in the paper. For example, on page 10, line 257, "HM" should be italicized. Please check the whole paper and correct similar mistakes carefully.

3.The format of the references is not uniform in the paper. For example, in references 19 and 32, "IEEE" is lowercase while it is uppercase in other references. The authors should modify the format of references carefully.

Reviewer #4: The authors have revised the manuscript. The overall structure of the paper is clear. The simulation results show that the proposed scheme is superior to the baseline approaches. To make this paper accepted, I think relevant content still need to be improved, which is shown below.

1. This paper involves the derivation of many formulas, but some derivations are too brief, such as page 9. It is suggested to add intermediate steps appropriately.

2. In the last part of the figure, it's best to remove the outer border of the picture to make the picture cleaner.

3. The format of the references is not uniform in the paper. The authors should modify the format of references carefully.

4. There are some punctuation mistakes. For example, when introducing contribution points on page 3, the first point ends with a comma and the other two points end with a full stop, hoping that the author can correct similar mistakes.

7. PLOS authors have the option to publish the peer review history of their article (what does this mean?). If published, this will include your full peer review and any attached files.

Reviewer #1: No

Reviewer #3: No

Reviewer #4: No

---

## [Author Response · Author response to Decision Letter 1]

13 Jun 2022

Dear Editor,

Thank you for allowing a resubmission of our manuscript, with an opportunity to address the reviewers’ comments.

We are uploading:

(1) Our point-by-point response to the comments (below) (response to reviewers). 

(2) An updated manuscript with Red Font Colour indicating changes, and 

(3) A clean updated manuscript without highlights.

(4) An image file (File of figures).

Best regards,

Dr. Khaled H Almotairi

Assistant Professor

Email: khmotairi@uqu.edu.sa

REVIEWER 1- COMMENTS

Comment 1

Reviewer #1: The paper has been improved. All the previous suggestions have been well considered and thoroughly implemented in this revised paper. Hence, this paper could be accepted for publication now.

Response to comment 1

Thank you very much for recommending acceptance of the paper. In addition, we would like to mention our sincere appreciation for your efforts in improving the manuscript.

REVIEWER 3- COMMENTS

Comment 1

Reviewer #3: After the revision of the original version, the following problems still exist in this paper:

1. This paper involves many algorithms, but the part of the derivation is too concise. I think it is necessary to explain the algorithms properly, which can help readers better understand them. It is suggested to give detailed and clear explanation.

Response to comment 1

Thank you for the valuable comments. To handle your suggestion, a detailed and comprehensive explanation of the algorithms have been provided (see Pages 11, 13 and 14). In addition, some of the steps required in the derivations have been added (see Page 9)

Comment 2

2.There are some wrong fonts in the paper. For example, on page 10, line 257, "HM" should be italicized. Please check the whole paper and correct similar mistakes carefully.

Response to comment 2

Thank you for the observation, “HM” and other variable terms have been corrected throughout the manuscript (see Pages 10 and 11).

Comment 3

3.The format of the references is not uniform in the paper. For example, in references 19 and 32, "IEEE" is lowercase while it is uppercase in other references. The authors should modify the format of references carefully.

Response to comment 3

We appreciate your comments, the whole of the reference section has been corrected in order to be uniform in terms of format. For example, see reference 19 and 32, thank you.

REVIEWER 4- COMMENTS

Comment 1

Reviewer #4: The authors have revised the manuscript. The overall structure of the paper is clear. The simulation results show that the proposed scheme is superior to the baseline approaches. To make this paper accepted, I think relevant content still need to be improved, which is shown below.

1. This paper involves the derivation of many formulas, but some derivations are too brief, such as page 9. It is suggested to add intermediate steps appropriately.

Response to comment 1

Thank you very much for the important comments. To acknowledge this comment, the required intermediate steps for derivation of the formulas have been included. For example, see Pages 8 and 9 of the manuscript.

Comment 2

2. In the last part of the figure, it's best to remove the outer border of the picture to make the picture cleaner.

Response to comment 2

We appreciate the precise comment, the outer border of the figure has been removed to make the figure clearer (see all figures in the manuscript).

Comment 3

3. The format of the references is not uniform in the paper. The authors should modify the format of references carefully.

Response to comment 3

Thank you, the whole of the reference has been formatted and corrected in order to be uniform in the whole of the manuscript (see Pages 25-27). 

Comment 4

4. There are some punctuation mistakes. For example, when introducing contribution points on page 3, the first point ends with a comma and the other two points end with a full stop, hoping that the author can correct similar mistakes.

Response to comment 4

We appreciate your observation; all punctuation mistakes have corrected throughout the manuscript. For example, see Page 3 of the manuscript.

---

## [Decision Letter · Decision Letter 2]

16 Aug 2022

An improved multipath video data communication in a vehicular delay-tolerant network

PONE-D-21-30574R2

Dear Dr. Almotairi,

We’re pleased to inform you that your manuscript has been judged scientifically suitable for publication and will be formally accepted for publication once it meets all outstanding technical requirements.

Kind regards,

Anand Nayyar, Ph.D.

Academic Editor

PLOS ONE

Additional Editor Comments (optional):

The Revised Paper stands Accepted with no further revisions.

Reviewers' comments:

Reviewer's Responses to Questions

**Comments to the Author**

1. If the authors have adequately addressed your comments raised in a previous round of review and you feel that this manuscript is now acceptable for publication, you may indicate that here to bypass the “Comments to the Author” section, enter your conflict of interest statement in the “Confidential to Editor” section, and submit your "Accept" recommendation.

Reviewer #1: All comments have been addressed

Reviewer #3: (No Response)

2. Is the manuscript technically sound, and do the data support the conclusions?

Reviewer #1: Yes

Reviewer #3: (No Response)

3. Has the statistical analysis been performed appropriately and rigorously? 

Reviewer #1: Yes

Reviewer #3: (No Response)

4. Have the authors made all data underlying the findings in their manuscript fully available?

Reviewer #1: Yes

Reviewer #3: (No Response)

5. Is the manuscript presented in an intelligible fashion and written in standard English?

Reviewer #1: Yes

Reviewer #3: (No Response)

6. Review Comments to the Author

Reviewer #1: I have no further comments. My previous comments and suggestions have been addressed in last version. This paper can be accepted for publication now.

Reviewer #3: I think the paper has been revised well. This paper is accpectable for this journal. So, I suggest to accpect the paper.

7. PLOS authors have the option to publish the peer review history of their article (what does this mean?). If published, this will include your full peer review and any attached files.

Reviewer #1: No

Reviewer #3: No

---

## [Editor Report · Acceptance letter]

8 Sep 2022

PONE-D-21-30574R2 

An improved multipath video data communication in a vehicular delay-tolerant network 

Dear Dr. Almotairi:

I'm pleased to inform you that your manuscript has been deemed suitable for publication in PLOS ONE. Congratulations! Your manuscript is now with our production department. 

Kind regards, 

on behalf of

Dr. Anand Nayyar 

Academic Editor

PLOS ONE